# Comparing the Effectiveness of Cognitive Rehabilitation and Binaural Beats on Craving and Comorbidities of Sexual Hyperactivity: A Pilot, Exploratory Quasi-Experimental Study

**DOI:** 10.3390/healthcare12111116

**Published:** 2024-05-30

**Authors:** Zeinab Mousavi, Mohammad Hossein Samanipour, Hamed Zarei, Payman Hassani Abharian, Halil İbrahim Ceylan, Nicola Luigi Bragazzi

**Affiliations:** 1Department of Psychology, Faculty of Human Science, Islamic Azad University, Varamin 1777613651, Iran; saharmousavi444@gmail.com; 2Department of Sport Science, Imam Khomeini International University, Qazvin 3414896818, Iran; samani.mh@gmail.com; 3Department of Biology, Central Tehran Branch, Islamic Azad University, Varamin 1777613651, Iran; h.zarei@iautmu.ac.ir; 4Department of Cognitive Psychology and Cognitive Rehabilitation, Institute For Cognitive Science Studies (IRICSS), Tehran 1658344575, Iran; abharian@icss.ac.ir; 5Physical Education and Sports Teaching Department, Faculty of Kazim Karabekir Education, Atatürk University, 25030 Erzurum, Turkey; 6Laboratory for Industrial and Applied Mathematics (LIAM), Department of Mathematics and Statistics, York University, Toronto, ON M3J 1P3, Canada; 7Human Nutrition Unit (HNU), Department of Food and Drugs, Medical School, University of Parma, 43125 Parma, Italy

**Keywords:** sexual hyperactivity, cognitive treatment, sexual disorders, cognitive function, anxiety, depression

## Abstract

Sexual hyperactivity, often linked with substantial psychological and social disturbances, remains under-researched, particularly in contexts like Iran where cultural and social norms may influence the reporting and treatment of such conditions. This study explores the therapeutic potential of cognitive rehabilitation (CR) and binaural beats (BB) in addressing this issue. The primary objective was to compare the effectiveness of CR and BB in reducing symptoms of sexual hyperactivity and associated comorbid conditions, with a focus on fluctuations in sexual desire and overall mental health. Utilizing a quasi-experimental design, the study involved pretest, posttest, and follow-up assessments to evaluate the interventions’ impacts. Recruitment through social media yielded 45 participants from a larger pool, who were then assigned to either the CR group, the BB group, or a control group. The CR and BB interventions were administered over a period of 10 sessions, each lasting 20 min, 3 times a week. Significant improvements were observed in both intervention groups compared to the control group. The CR group showed a marked reduction in Sexual Addiction Screening Test (SAST) scores from an initial average of 24.87 to 6.80 at follow-up, indicating a reduction in symptoms of sexual hyperactivity. The BB group also showed improvement, with SAST scores decreasing from 19.93 to 9.57. In terms of mental health comorbidities, the Depression, Anxiety, and Stress Scale (DASS-21) scores decreased notably in the CR group from a baseline of 8.53 to 3.07 post-intervention, and in the BB group from 10.33 to 5.80. Both interventions showed similar effectiveness in reducing anxiety and stress, with no statistically significant differences between the groups for most of the outcomes studied, affirming their potential for clinical application.

## 1. Introduction

Hyperactive sexuality, known also as hypersexual disorder (HSD) [1] or problematic hypersexuality [1], has become a significant concern globally, marked by individuals’ struggles to control their excessive focus on sexual thoughts, impulses, and activities, which often interfere with their daily life and mental well-being [1,2]. This condition, also referred to as sexual addiction (SA), is characterized by persistent, unsuccessful efforts to limit the amount of time spent on sexual fantasies, urges, and behaviors, typically triggered by everyday emotional states or stressful situations [3]. The term SA is gaining acceptance within the psychiatric and psychological communities due to the significant social and personal challenges it poses for many individuals [3].

Despite HSD/SA not being officially classified in the DSM-5, the clinical and scientific communities acknowledge the existence of hypersexuality as a severe and uncontrollable form of sexual behavior, even if some scholars have criticized the theoretical framework as harmful and unnecessary [4] or not properly conceptualized and operationalized [5], thus urging for a more cautious approach. On the other hand, despite the lack of a standardized terminology (dysregulated sexuality, dysfunctional sexuality, excessive sexuality, or sexual compulsivity, among others), there is a consensus on the need to better understand its underlying causes and develop evidence-based treatments [6,7]. In 2019, compulsive sexual behavior disorder (CSBD) was officially acknowledged as a new diagnostic construct and introduced among the impulse-control disorders of the International Classification of Diseases 11th Revision (ICD-11) edited and published by the World Health Organization (WHO) [6]. Individuals with CBSD exhibit poorly controlled impulsive sexual behaviors, along with obsessive thoughts, including sexual fantasization [8].

The prevalence of broadly defined sexual hyperactivity is noticeable, with studies showing around 12% of men and 7% of women experiencing these issues, while other research indicates a prevalence rate of 3% to 6% in the general population [9,10,11]. Such discrepancies may reflect the heterogeneity of the populations recruited as well as methodological differences in the definition and assessment of the construct under study [12,13,14], which appears to be complex and exhibits a latent dimensionality [15,16].

The academic interest in sexual hyperactivity has notably increased, leading to the development of various diagnostic tools [17], like the Hypersexual Disorder Screening Inventory (HDSI) [18], the Hypersexual Behavior Inventory (HBI) [19,20], or the Sexual Compulsivity Scale (SCS) [20], among others, aimed at identifying and measuring the extent of hypersexuality-related disorders [17]. These can have severe consequences [16], including an increased likelihood of committing sexual offenses and misconducts [21] and using pornography in ways that reinforce negative attitudes towards women [22,23,24].

On a broader scale, compulsively engaging in sexual activities is seen as a way for individuals to escape uncomfortable, unresolved, and distressing emotions. The disorder’s negative impact includes potential job loss, legal problems, social isolation, increased risk of divorce, and health risks associated with sexually transmitted infections [25], among others. Given these challenges, it is crucial for mental health professionals to recognize sexual hyperactivity as a significant indicator of deeper emotional and psychological issues.

Discussions among experts focus on the core principles of diagnosis, non-pharmacological and pharmacological treatments, highlighting the importance of standardized therapeutic approaches [26,27]. Like other behavioral addictions, treating hypersexuality requires a personalized approach that considers the disorder’s unique aspects and includes treatment for co-occurring psychiatric and physical conditions, ensuring a comprehensive treatment strategy.

Epidemiological and clinical research suggests that impaired cognition may predispose individuals to impulsivity, addiction tendencies, and increased sexual desire [28,29,30]. Addressing these cognitive issues through cognitive rehabilitation and binaural beat techniques may offer new avenues for treatment.

Cognitive deficits, often linked to various psychiatric conditions, play a crucial role in sexual dysfunction and are a key focus of comprehensive rehabilitation programs that include social skills training, psychological interventions, and counseling for sexual dysfunction [31,32]. Cognitive rehabilitation (CR), focusing on improving attention, memory, executive functions, social cognition, and metacognition, is vital for enhancing cognitive functioning in individuals with sexual hyperactivity [31,32]. Recent studies highlight the potential of binaural beats (BB), an auditory phenomenon occurring when two tones of different frequencies, presented separately to each ear, elicit the sensation of a third tone oscillating at the difference frequency of the two tones. This technology seems to entrain the brain by inducing changes in brain activity through auditory stimuli, potentially improving cognitive function and reducing symptoms related to anxiety, sleep disorders, mood, and difficulties with concentration [33,34,35,36,37]. This exploration of CR and BB as promising treatments for psychiatric and psychological conditions related to executive functions emphasizes their effectiveness in improving attention, selective attention, inhibitory control, working memory, and sustained attention, which are crucial for managing sexual hyperactivity and its associated psychological distress.

However, there is a noticeable gap in the literature, especially regarding the comparative effectiveness of these interventions for treating symptoms of sexual hyperactivity and related conditions, particularly within specific populations like Iranian males aged from 20 to 40 years.

This study aims to fill this gap by evaluating and comparing the efficacy of CR- and BB-based interventions in reducing craving tendencies and associated comorbid conditions among this demographic, ultimately contributing to a better understanding of these therapeutic options in the context of sexual hyperactivity.

## 2. Materials and Methods

### 2.1. Study Design

The current research employed a quasi-experimental methodology, implementing a pretest–posttest follow-up test design framework. The participant recruitment phase was initiated through direct outreach and communication, targeting a pool of 1500 men via messages disseminated across various social media platforms. This resulted in 250 individuals expressing their willingness to engage in the study. The administration of preliminary questionnaires related to sexual desire to these volunteers yielded complete responses from 100 participants. Further scrutiny of these responses enabled the identification of 60 participants who met the diagnostic criteria for CSBD. Nonetheless, the challenges imposed by the “Coronavirus Disease 2019” (COVID-19) pandemic, alongside other unforeseen factors, resulted in a final sample of 45 men who consistently participated throughout the entire duration of the study. Characteristics of the entire sample recruited are presented in Table 1.

Subsequent to their arrival at the laboratory, an in-depth diagnostic interview focusing on sexual hyperactivity was conducted with each participant. It is important to note that the study’s methodological rigor is supported by previous research [38], which suggests that a sample size of 15 to 20 individuals per group is considered methodologically sound. Adhering to this guideline, 45 eligible participants were selected to partake in the study. These participants were then allocated into three distinct groups: the CR group (comprising 15 individuals), the BB-based intervention group (also consisting of 15 individuals), and a control group (with 15 individuals).

The inclusion and exclusion criteria were stringently defined to ensure the study’s integrity and soundness. Participants were required to be free from any addiction to narcotics, devoid of any significant physical illnesses, not under any psychiatric medication, aged between 20 and 40, and willing to provide written, informed consent. Conversely, the exclusion criteria included individuals with non-heterosexual orientation, with sexual paraphilias, those on psychiatric medications, individuals with a diagnosis of major depressive disorder or anxiety disorders, those with a history of brain trauma or epilepsy, and individuals with a history of or current heart diseases.

### 2.2. Study’s Procedures

This study’s intervention phase was meticulously designed. The first experimental group underwent a CR program, spanning 10 sessions, each lasting 20 min, and conducted 3 times a week. As reported in a previous study [29], CR comprises a collection of non-pharmacological approaches aimed at promoting consistency, regulation, and mitigation of cognitive impairments observed in certain individuals. The second experimental group was exposed to a BB-based intervention, utilizing frequencies specifically set at 40 Hz within the gamma range, entailing a 300 Hz frequency for the right ear and a 340 Hz frequency for the left ear, generated using Gnaural (available at https://gnaural.sourceforge.net/) and Adobe Audition software (version 3.0). Participants in this group underwent 10 sessions, each 20 min long, occurring three times a week, and facilitated through standard headphones. The control group, in contrast, was exposed solely to instrumental, non-verbal music with frequencies set from 300 to 300 Hz, with these auditory sessions also lasting 20 min and conducted 3 times a week over a month. Post-test evaluations were carried out following the intervention’s conclusion, with a follow-up assessment conducted two months later to gauge the interventions’ sustained effects on the targeted variables.

### 2.3. Study Instruments and Questionnaires Section

Several questionnaires and tasks were employed to measure various outcomes. These are described in detail in the following sub-sections.

#### 2.3.1. Carnes et al.’s Sexual Addiction Screening Test

The Carnes et al.’s Sexual Addiction Screening Test (SAST), comprising 45 binary (Yes/No) questions, was administered. These questions cover a range of behaviors and thoughts related to sexual activity, including the following: (i) the frequency and intensity of sexual thoughts and behaviors; (ii) the emotional consequences of these sexual behaviors; (iii) the impact of sexual behavior on relationships, work, and social life; and (iv) the use of sexual behavior as a response to stress or emotional discomfort. This tool plays a critical role in the broader context of understanding and treating hypersexuality, providing a structured approach to identifying potentially harmful patterns that may require professional attention. The reliability of this questionnaire has been thoroughly evaluated, with Cronbach’s alpha coefficients indicating strong internal consistency across different demographics [39]. Additionally, the questionnaire’s validity for the Iranian population was confirmed through a study by Zahedian et al., which reported satisfactory psychometric properties [40].

#### 2.3.2. Depression, Anxiety, and Stress 21-Item Questionnaire

The Depression, Anxiety, and Stress Questionnaire (DASS-21), designed to measure the psychological constructs of depression, anxiety, and stress, was also administered. This shortened version of the original 42-item self-report scale maintains the core characteristics of the full version and is widely used for its brevity and effectiveness in research and clinical settings. Each item is rated on a 4-point Likert scale, ranging from 0 (“it did not apply to me at all”) to 3 (“it applied to me very much or most of the time”). The DASS-21 is divided into three distinct sections, each containing seven items that assess related emotional states. The depression scale measures various aspects such as feelings of sadness (dysphoria), hopelessness, low self-worth, disinterest in life, inability to feel pleasure (anhedonia), and psycho-physiological signs of depression like fatigue or inertia. The anxiety scale evaluates symptoms like autonomic arousal, effects on muscles, situational anxiety, and the general feeling of anxiousness. The stress scale focuses on detecting chronic, non-specific arousal, capturing an individual’s difficulty in relaxing, tendency to be nervous, and quickness to feel upset or agitated, as well as levels of irritability and impatience. To determine scores for depression, anxiety, and stress, the responses to the relevant items on each scale are summed. The existing scholarly literature indicates that the DASS-21 can capture a significant portion of the variability associated with these psychological constructs, with high internal consistency demonstrated by the Cronbach’s alpha coefficients for stress, depression, and anxiety factors [41].

#### 2.3.3. Stop-Signal Task

The Stop-Signal Task (SST) is a well-established method for assessing response inhibition which utilizes visual stimuli and requires participants to make quick responses using a computer keyboard. The task’s design ensures that the stop-signal delay is adjusted based on individual performance, allowing for a tailored assessment of each participant’s inhibitory control capabilities [42].

#### 2.3.4. The Go/No-Go Task

The Go/No-Go task, rooted in the Go-No-Go paradigm, is a foundational tool in cognitive psychology and neuroscience, providing valuable insights into the mechanisms underlying human behavior and mental health disorders. It is renowned for its ability to evaluate response inhibition, behavioral restraint, and self-control. This task presents stimuli in a randomized manner, with participants required to respond to ‘go’ stimuli while withholding their response to ‘no-go’ stimuli. The task’s design minimizes anticipatory responses and emphasizes the dominance of the ‘go’ response, with performance measured by the minimization of errors, which can be categorized into commission errors (occurring when a participant responds to a ‘no-go’ stimulus, indicating a lapse in inhibitory control) and omission errors (occurring when a participant fails to respond to a ‘go’ stimulus, possibly indicating issues with attention or slower processing speed) [43,44].

### 2.4. Statistical Analyses

Statistical analyses included descriptive statistics (means and standard deviations), chi-squared test, analysis of variance (ANOVA), and repeated measures ANOVA (rmANOVA), after verifying underlying assumptions (sphericity, normality, independence, absence of multicollinearity and of significant outliers, homogeneity of variances, and stationarity). A significance level of *p* < 0.05 was established, with the exception of multiple comparisons and testing when Tukey’s correction was applied. All statistical analyses were carried out in the open-source R environment (R version 4.2.3).

## 3. Results

### 3.1. Population Recruited

The three groups did not differ in terms of mean age (BB group age: 32.27 ± 6.93 years, range 20–40; CR group age: 32.13 ± 5.07 years, range 21–39; control group age: 32.67 ± 6.26 years, range 22–40; F = 0.031, *p* = 0.970, $\eta_{p}^{2}=0.001)$ as well as in terms of marital status (chi-square statistic = 2.14, *p* = 0.343), socio-economic status (chi-square statistic = 2.70, *p* = 0.609), and employment status (chi-square statistic = 1.68, *p* = 0.431).

### 3.2. The Sex Addiction Screening Test

Concerning the study outcomes, and more specifically SAST, both time (F = 207.0, *p* < 0.001) and intervention (F = 144.0, *p* < 0.001) had a significant impact, as well as their interaction (F = 53.9, *p* < 0.001). At the start, the CR group scored the highest (24.87 ± 2.10) compared to the other two groups, indicating more severe symptoms of hypersexuality. This group showed a dramatic decrease in SAST scores from baseline to post-intervention (4.6 ± 1.35), though the scores increased again by the follow-up (6.80 ± 3.69) but remained considerably lower than their starting point. Altogether, this suggests that the intervention was highly effective initially, even if its effects were only partly long-lasting. The BB group, starting at a lower baseline than the CR group (19.93 ± 1.53), also registered a significant reduction in SAST scores from baseline to post-intervention (7.53 ± 1.64). Their scores rose slightly at follow-up, reaching values comparable with those of the CR group (9.57 ± 5.17). Finally, the control group, which received no specific intervention, started with SAST scores similar to those of the BB group (21.53 ± 1.73), and showed only slight changes, with a minor decrease from baseline to post-intervention (21.87 ± 1.64) and another small decrease by the follow-up (18.73 ± 3.75). Overall, at the rm-ANOVA, both intervention approaches—BB and CR—initially reduced hypersexuality symptoms effectively. Even if the latter reduced the symptoms to a greater extent, they demonstrated comparable effects at the follow-up.

At the post hoc analysis, the baseline differed from both post-intervention (mean difference = 10.778, SE = 0.376, df = 42, t = 28.629, *p* < 0.001) and follow-up (mean difference = 10.411, SE = 0.701, df = 42, t = 14.853, *p* < 0.001), while there was no difference between post-intervention and follow-up (mean difference = −0.367, SE = 0.673, df = 42, t = −0.545, *p* = 0.850). The BB and the CR groups showed a minimal mean difference of 0.256 (SE = 0.578), indicating negligible differences between these interventions in terms of the measured outcome (t = 0.442, df = 42, *p* = 0.898). Conversely, when the BB group is compared to the control group, a significant mean difference of −8.367 could be computed, which suggests that the control group underperformed substantially (t = −14.485, df = 42, *p* < 0.001). Similarly, the comparison between the CR and the control groups revealed a mean difference of −8.622 (t = −14.927, df = 42, *p* < 0.001) (Figure 1).

### 3.3. Depression

Concerning the DASS-21 (Figure 2), in terms of depression, both time (F = 49.5, *p* < 0.001) and intervention (F = 69.1, *p* < 0.001) were significant, as well as their interplay (F = 20.3, *p* < 0.001). Baseline differed from post-intervention (mean difference = 3.289, SE = 0.296, df = 42.0, t = 11.12, *p* < 0.001) and follow-up (mean difference = 2.378, SE = 0.378, df = 42.0, t = 6.29, *p* < 0.001). Post-intervention differed as well from follow-up (mean difference = −0.911, SE = 0.345, df = 42.0, t = −2.64, *p* = 0.031). More in detail, the CR overperformed with respect to the BB and the control groups. Baseline depression levels were comparable in all three groups (BB group: 10.33 ± 1.496; CR group: 8.53 ± 1.846; control group: 10.27 ± 1.163), decreasing significantly in the CR group (3.07 ± 0.884) and to a less extent in the BB group (5.80 ± 1.656), with a slight increase in the control group (10.40 ± 1.765). At the follow-up, there was an increase in all three groups (BB group: 5.07 ± 1.534; CR group: 6.00 ± 1.732; control group: 10.93 ± 2.865). At the post hoc analysis, in the comparison between the BB and the CR groups, there was a mean difference of 1.20, indicating that the BB group underperformed compared with the CR group. This difference is statistically significant with a standard error of 0.412 (df = 42, t = 2.91, *p* = 0.016). When comparing the BB group to the control group, the mean difference was computed at −3.47, showing the beneficial impact of the BB intervention (t = −8.41, *p* < 0.001). Finally, the CR group scored 4.67 units lower on average compared to the control group (t = −11.32, *p* < 0.001).

### 3.4. Anxiety

Regarding anxiety, both time (F = 36.9, *p* < 0.001) and intervention (F = 42.5, *p* < 0.001) were impactful, as well as their interaction (F = 21.9, *p* < 0.001). All three timepoints differed from each other: baseline from post-intervention (mean difference = 1.47, SE = 0.462, df = 42, t = 3.19, *p* = 0.007) and follow-up (mean difference = 3.47, SE = 0.368, df = 42, t = 9.43, *p* < 0.001), as well as post-intervention from follow-up (mean difference = 2.00, SE = 0.38, df = 42, t = 5.26, *p* < 0.001). Baseline anxiety levels were similar across the three groups (BB group: 10.15 ± 2.818; CR group: 11.00 ± 2.449; control group: 9.53 ± 1.36) and decreased after the intervention in the BB (8.33 ± 1.397) and in the CR groups (6.53 ± 1.685), while they increased in the control group (11.40 ± 1.957). These values further decreased at the follow-up (achieving scores of 5.40 ± 2.063 and 3.80 ± 1.935 in the BB and in the CR groups, respectively), with only a slight decrease in the control group (11.07 ± 0.884). At the post hoc analysis, there was a mean difference of 0.85 between the BB and the CR groups (SE = 0.403, df = 42, t = 2.11, *p* = 0.099). Both the BB (mean difference = −2.704, t = −6.71, *p* < 0.001) and the CR groups (mean difference = −3.556, t = −8.83, *p* < 0.001) significantly differed from the control group.

### 3.5. Stress

Concerning stress, both time (F = 146.0, *p* < 0.001) and intervention (F = 62.9, *p* < 0.001) were impactful, as well as their interplay (F = 16.8, *p* < 0.001). In terms of time, baseline and post-intervention differed significantly (mean difference = 3.511, SE = 0.245, df = 42, t = 14.32, *p* < 0.001), as well as baseline and follow-up (mean difference = 4.467, SE = 0.264, df = 42, t = 16.93, *p* < 0.001). Similarly, the difference between post-intervention and follow-up was significant (mean difference = 0.956, SE = 0.312, df = 42, t = 3.06, *p* = 0.011). Baseline stress levels were comparable in all three groups (BB group: 10.27 ± 1.710; CR group: 10.80 ± 1.207; control group: 12.07 ± 1.033). After the intervention, there was a substantial decrease in the BB (5.53 ± 1.846) and the CR groups (7.27 ± 1.438), as well as in the control group (9.80 ± 0.775). At the follow-up, scores further decreased in the BB (4.27 ± 2.219) and in the CR groups (4.80 ± 1.656), with a slight increase in the control group (10.67 ± 1.113). At the post hoc analysis, the BB group differed from the CR group only marginally (mean difference = −0.933, SE = 0.389, df = 42, t = −2.40, *p* = 0.053), while differed significantly from the control group (mean difference = −4.156, t = −10.69, *p* < 0.001). Similarly, the CR group differed from the control group in a statistically significant fashion (mean difference = −3.222, t = −8.29, *p* < 0.001).

### 3.6. Selective Reaction Time

Regarding selective reaction times (Figure 3), both time (F = 5.71, *p* = 0.005) and intervention (F = 13.6, *p* < 0.001) were significant, but not their interaction (F = 1.41, *p* = 0.238). Baseline differed from post-intervention (mean difference = 25.61, SE = 9.89, df = 42, t = 2.59, *p* = 0.034) and follow-up (mean difference = 18.88, SE = 6.63, df = 42, t = 2.85, *p* = 0.018), while there was no difference between post-intervention and follow-up (mean difference = −6.73, SE = 6.59, df = 42, t = −1.02, *p* = 0.567). Baseline values were comparable among all three groups (BB group: 588.53 ± 3.097 ms; CR group: 567.91 ± 76.395 ms; control group: 591.39 ± 1.309 ms), which decreased after the intervention (BB group: 552.51 ± 1.980 ms; CR group: 531.34 ± 78.137 ms; control group: 587.13 ± 5.963 ms) and remained stable at the follow-up (BB group: 551.30 ± 3.895 ms; CR group: 552.43 ± 3.950 ms; control group: 587.44 ± 3.604 ms). At the post hoc analysis, there was no difference between the BB and CR groups (mean difference = 13.6, SE = 7.41, df = 42, t = 1.83, *p* = 0.173). Both BB (mean difference = −24.5, t = −3.31, *p* = 0.005) and CR groups (mean difference = −38.1, t = −5.14, *p* < 0.001) differed from the control group.

### 3.7. Stop-Signal Task

Concerning the SST paradigm, the temporal trend of the latency to effectively stop an ongoing response, or, in other words, successfully complete its cancelation for the three study groups is pictorially shown in Figure 4. Both time (F = 61.4, *p* < 0.001) and intervention (F = 50.3, *p* < 0.001), as well as their interaction (F = 12.5, *p* < 0.001) were significant. In terms of time, baseline differed from post-intervention (mean difference = 29.62, SE = 2.95, df = 42, t = 10.026, *p* < 0.001) and follow-up (mean difference = 27.80, SE = 2.70, df = 42, t = 10.286, *p* < 0.001), while post-intervention did not differ from follow-up (mean difference = −1.81, SE = 3.30, df = 42, t = −0.549, *p* = 0.847). At the post hoc analysis, no difference could be found between the CR and BB groups (mean difference = −5.36, SE = 3.15, df = 42, t = −1.70, *p* = 0.216), while both the BB (mean difference = −29.61, t = −9.41, *p* < 0.001) and the CR groups (mean difference = −24.25, t = −7.71, *p* < 0.001) differed from the control group. Baseline stop-signal reaction times were comparable in the three groups (BB group: 287.55 ± 15.320 ms; CR group: 290.11 ± 2.854 ms; control group: 292.58 ± 2.149 ms), which decreased at the second time-point in all the groups (BB group: 250.53 ± 1.874 ms; CR group: 246.11 ± 28.022 ms; control group: 284.75 ± 10.314 ms). Final stop-signal reaction times were lowest in the BB group (239.77 ± 26.179 ms), followed by the CR group (257.71 ± 6.705 ms) and highest in the control group (289.35 ± 2.691 ms). All this suggests that both the BB and CR interventions may lead to slight improvements in reaction time that are maintained over time, as shown by the stable values from post-intervention to follow-up. The control group showed, instead, little change, implying that the interventions might be responsible for the improvements seen.

### 3.8. Stop-Signal Delay Time

Regarding stop-signal delay times, both time (F = 161.0, *p* < 0.001) and intervention (F = 176, *p* < 0.001) were significant, as well as their interaction (F = 54.2, *p* < 0.001). Baseline differed significantly from post-intervention (mean difference = −13.41, SE = 0.83, df = 42, t = −16.17, *p* < 0.001) and follow-up (mean difference = −13.39, SE = 0.81, DF = 42, t = −16.60, *p* < 0.001), while there was no difference between post-intervention and follow-up (mean difference = 0.01, SE = 0.94, df = 42, t = 0.01, *p* = 1.000). Baseline stop-signal delay times were comparable in all three groups (BB group: 283.72 ± 2.767 ms; CR group: 289.93 ± 2.154 ms; control group: 288.93 ± 3.120 ms), which increased after the intervention in the BB (308.75 ± 2.694 ms) and the CR groups (307.13 ± 6.193 ms), with only a slight decrease in the control group (286.93 ± 2.657 ms). Finally, stop-signal delay times were found to be stable at the follow-up in all three groups (BB group: 304.83 ± 6.374 ms; CR group: 310.07 ± 4.395 ms; control group: 287.87 ± 3.118 ms). At the post hoc analysis, the BB group differed from both the CR (mean difference = −3.28, SE = 0.808, df = 42, t = −4.06, *p* < 0.001) and the control (mean difference = 11.19, t = 13.85, *p* < 0.001) groups. The CR group differed as well from the control group (mean difference = 14.47, t = 17.92, *p* < 0.001).

## 4. Discussion

The findings of this investigation highlight the significant improvements in sexual hyperactivity brought about by both BB and CR interventions. Moreover, the post hoc analysis revealed no significant differences between the efficacy of BB and CR in mitigating symptoms of sexual hyperactivity for most of the outcomes under study. This observation is in line with the perspectives shared by Love et al. [45], who emphasized the detrimental effects of hypersexuality, particularly pornography addiction, on neuro-cognitive functions, including cognitive flexibility. They suggested that such addictions are associated with notable impairments in cognitive processes.

BB and CR therapies are rooted in the exploration of perceptual experiences and cognitive processes, offering promising avenues for addressing hyperactive sexual behaviors. Despite the limited focus on hypersexuality within Iranian research [46,47], and the broader international context, the findings from this study align with the existing literature, highlighting a gap in the comparative analysis of these therapeutic approaches’ effectiveness in SA treatment. BB technology operates by delivering two distinct auditory signals with different frequencies to each ear, leading to the perception of a third tone that represents the frequency difference between the two inputs [48,49].

This phenomenon has been explored in various frequency bands, demonstrating potential benefits such as improved memory and attention within the beta frequency band and enhanced relaxation states within the alpha frequency band [48,49]. The potential link between the behavioral effects observed in this study, particularly with 40 Hz binaural beats, and changes in theta and gamma wave activities, suggests a complex interaction where increased gamma activity, indicative of physiological activation, may result from the modulation of theta base activity or its balance.

This hypothesis seems to be supported by studies showing that gamma wave activity increases during emotional processing and engagement. For instance, the investigation by Khorramabadi and Asadi [50] highlights the benefits of music therapy and auditory phenomena, including BB, in enhancing mental and social well-being by inducing alpha brainwave activity and promoting relaxation. Alipoor et al. [51] also found that BB-induced brainwave synchronization effectively reduced anxiety levels, aligning with the findings of Becher et al. [52] and others [53,54,55], who observed improvements in cognitive functions like selective attention following binaural beat exposure at gamma frequencies. Similarly, CR encompasses restorative and compensatory approaches, focusing on cognitive exercises and strategic methods to overcome functional impairments, respectively [56,57].

This holistic approach can lead to improvements in memory, response inhibition, and cognitive organization, contributing to anxiety reduction as evidenced in both post-test and follow-up phases of this study. The use of a 40 Hz BB, corresponding to gamma rhythm, in previous studies [58,59,60,61] has shown positive effects on stress reduction and cognitive function enhancement, consistent with the findings of this investigation.

The rhythmic influence on brain dynamics, particularly through activities like drumming, underscores the potential of BB in modulating consciousness states and frontal cortical activity [58,59,60,61]. The anxiolytic effects of BB, particularly at 40 Hz, may involve changes in frontal EEG synchrony, with implications for anxiety disorders. This reduction in anxiety might be linked to the modulation of dopamine levels within the amygdala [62,63], indicating a complex interaction between BB and neurochemical processes.

CR’s effectiveness in reducing anxiety is attributed to its impact on processing speed, cognitive flexibility, and memory functions, along with modulating activity in the parietal cortex within alpha and theta frequency bands. The principle of brain plasticity underlies the lasting changes induced by CR, suggesting structural and functional adaptations in response to cognitive exercises [64].

## 5. Study’s Strengths and Limitations

The present study has several strengths, including its comprehensive and methodologically rigorous approach, coupled with the use of validated measurement tools and innovative interventions, underscoring the study’s potential to contribute valuable insights into the effective management of hypersexuality-related symptoms and associated conditions.

On the other hand, this study’s limitations include its study design (quasi-experimental trial). Future research should be designed as randomized controlled trials (RCTs), or when randomization is not feasible, it should make use of advanced techniques, such as propensity-score matching, which enable to balance observational data, reducing confounding and bias when estimating the effects of a treatment, and improving causal inference by mimicking a randomized experimental design. Other shortcomings are the small sample size employed and the very low response rate to the initial outreach, as well as the study’s focus on male only participants and the impact of the COVID-19 pandemic on the research methodology. Future investigations should include EEG assessments, extend to female participants, and explore the vulnerability of teenagers and young adults to sexual issues and addiction. The promising results from BB and CR interventions advocate for their application in clinical settings, offering valuable tools for psychologists and psychiatrists in treating sexual hyperactivity and its associated conditions. As mentioned above, some suggest that that CR methodologies may offer superior ease of application and cognitive domain specificity tailored to individual mental frameworks and neurological facets. In light of our study focusing on the effectiveness of dual interventions, it could be posited that the combination of these two approaches may yield greater efficacy for individuals with hypersexual tendencies compared to either one applied in isolation.

## 6. Conclusions

This research underscores the potential of CR and BB interventions as effective strategies for psychiatrists and clinicians in the treatment of sexual hyperactivity disorders such as SA and HSD. The outcomes of these interventions highlight their capacity to significantly mitigate sexual cravings and their associated comorbidities, offering a new paradigm in therapeutic approaches within clinical settings. Cognitive rehabilitation, in particular, has been identified as a promising method for bolstering executive functions and enhancing overall cognitive abilities. This approach systematically targets and strengthens the cognitive processes that are often compromised in individuals suffering from SA and HSD, thereby facilitating improved self-regulation and behavioral control. The binaural beat technique, known for its distinctive impact on the frontal cortex, stands out for its ability to modulate brain activity more effectively than other cortical stimulation methods. This unique characteristic suggests that binaural beats could play a crucial role in addressing the neurological underpinnings of SA and HSD, particularly in individuals experiencing intense sexual cravings.

The influence of BB on the frontal cortex, an area integral to decision-making and impulse control, indicates that this intervention could directly affect the core cognitive challenges associated with these disorders. Given these insights, it becomes evident that integrating CR and BB therapies into the therapeutic repertoire for hypersexuality could offer a multifaceted approach, targeting both the cognitive and neurophysiological dimensions of these conditions. This holistic strategy not only addresses the symptomatic manifestations of sexual hyperactivity but also aims to rectify the underlying cognitive and brain function irregularities, paving the way for a more comprehensive and effective treatment paradigm for individuals grappling with these complex disorders.

However, given the limitations of the present pilot, exploratory study, further research in the field is urgently warranted.

## Figures and Tables

**Figure 1 healthcare-12-01116-f001:**
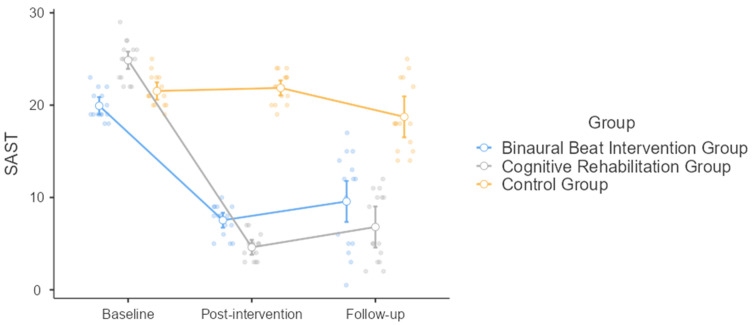
Sexual Addiction Screening Test (SAST) scores at the baseline, post-intervention, and at the follow-up for the three study groups (binaural beat intervention group; cognitive rehabilitation group; and control group).

**Figure 2 healthcare-12-01116-f002:**
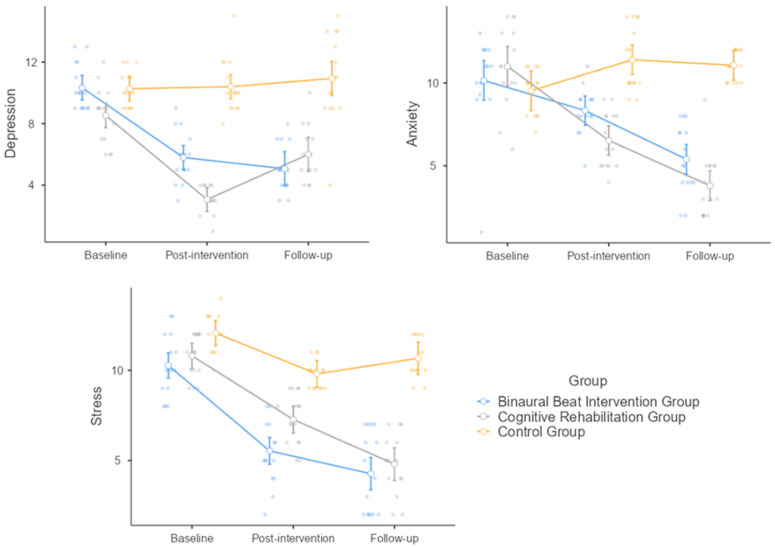
Depression, anxiety, and stress scale (DASS-21) scores at the baseline, post-intervention, and at the follow-up for the three study groups (binaural beat intervention group; cognitive rehabilitation group; and control group).

**Figure 3 healthcare-12-01116-f003:**
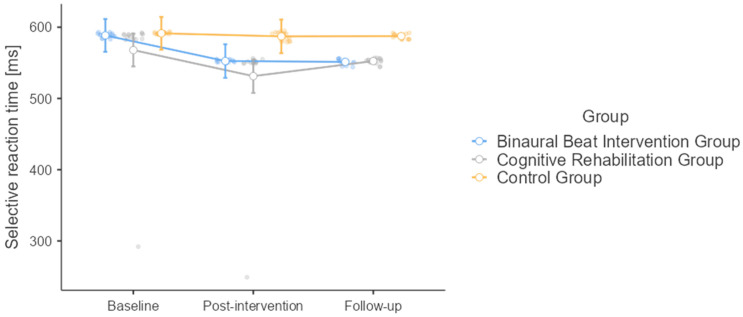
Selective reaction times at the baseline, post-intervention, and at the follow-up for the three study groups (binaural beat intervention group; cognitive rehabilitation group; and control group).

**Figure 4 healthcare-12-01116-f004:**
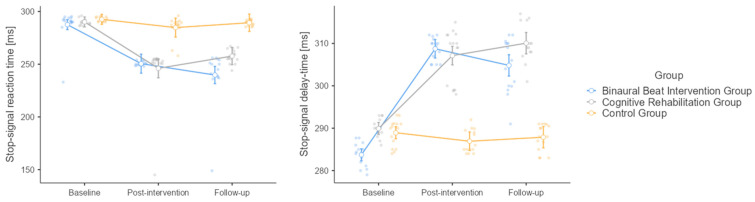
Stop-signal task related outcomes (stop-signal reaction time and stop-signal delay time) in the three study groups (binaural beat intervention group; cognitive rehabilitation group; control group).

**Table 1 healthcare-12-01116-t001:** Characteristics of the sample recruited.

Parameter	Value
Age	32.36 ± 6.00 years (range 20–40 years)
Socio-economic status	
Low	13 (28.9%)
Medium	23 (51.1%)
High	9 (20.0%)
Job status	
Employed	34 (75.6%)
Unemployed	11 (24.4%)
Marital status	
Married	24 (53.3%)
Not married	21 (46.7%)

## Data Availability

Data are available for research purposes upon reasonable request to the corresponding author.

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
