# Peer review of "Comparing the Effectiveness of Cognitive Rehabilitation and Binaural Beats on Craving and Comorbidities of Sexual Hyperactivity: A Pilot, Exploratory Quasi-Experimental Study"

_healthcare, 2024, doi:10.3390/healthcare12111116_

Round 1

Reviewer 1 Report

Comments and Suggestions for Authors

The study is well-conducted and the manuscript is generally well-written. Only some minor points would be improved.

1. The sample size of this study was small.

2. The process of intervention (duration and time interval) could be clarified.

3. The demographic data should be listed in Table 1.

Author Response

Dear reviewer, 

we have specified that the small sample size is one of the limitations of the study, as well as the low response rate (also due to the challenges of COVID-19). This limits the generalization of our findings and calls for caution when interpreting the findings. We have clarified the timing and duration of the interventions and we have now provided demographics of the sample recruited. 

Reviewer 1

The study is well-conducted and the manuscript is generally well-written. Only some minor points would be improved.

Response: Dear Reviewer, Thank you so much for your comments. We showed all improvements with yellow in the text.

  1. The sample size of this study was small.

Response: Dear Reviewer, Thank you so much for your comment. We have specified that the small sample size is one of the limitations of the study, as well as the low response rate (also due to the challenges of COVID-19). This limits the generalization of our findings and calls for caution when interpreting the findings.

  1. The process of intervention (duration and time interval) could be clarified.

Response:  We have clarified the timing and duration of the intervention in abstract and methods section clearly (Line 158-164)

  1. The demographic data should be listed in Table 1.

Response: Dear Reviewer, Thank you so much. We provided demographic data results at the first two paragraph of results section (Line 235-246)

Reviewer 2 Report

Comments and Suggestions for Authors

To the authors,

 The objective of this research aimed at clarifying the level of efficacy of different therapeutic approaches (Cognitive Rehabilitation and binaural beats) in addressing the symptoms of sexual hyperactivity and associated comorbid conditions, seems to us of interest, because of the important consequences of this problem at different levels (personal, family, health, ...).

In general, the paper is well written, I have only few comments that I hope could be helpful in improving the manuscript.

Introduction

 ·         Reading the introduction one confirms the use, in previous literature, of different terms (e.g., Sexual Addiction, Compulsive Sexual Disorder and Hyperactive Sexuality Disorder) to refer to the problem under analysis in this study. Therefore, we suggest that the authors refer to the lack of conceptual clarification in this regard and that they "bet" on a single term throughout the paper, justifying its use. Some articles such as those by Karila, L., Wery, A., Weinstein, A., Cottencin, O., Petit, A., Reynaud, M., & Billieux, J. (2014). Sexual addiction or hypersexual disorder: different terms for the same problem? A review of the literature. Current pharmaceutical design, 20(25), 4012-4020 and the one of Montgomery-Graham, S. (2017). Conceptualization and assessment of hypersexual disorder: A systematic review of the literature. Sexual Medicine Reviews, 5(2), 146-162, may be helpful.

 ·         In line 74, when it is stated that "Research suggests that cognitive impairments may predispose individuals to impulsivity, addiction tendencies, and increased sexual desire," can the authors specify what that research is?

 Materials and Methods

  • This section would be clearer and easier for the potential reader to follow if the authors structured it in different subsections: Procedure, participants, instruments and statistical analysis. Some aspects to consider:
    • Provide more information about the characteristics of the cognitive rehabilitation program that is applied in this research (line 127).
    • Include the authors' reference when presenting each of the assessment instruments (e.g., lines 140 and 146).
    • Provide more details about the characteristics of the Carnes Sex Addiction Screening Test (e.g., main dimensions that they assess). In addition, we suggest the inclusion of some items from this questionnaire to exemplify its content.
    • Indicate which Likert scale of measurement is used in the Depression, Anxiety, Stress Questionnaire (DASS).

o   Include a section explaining, taking into account the objectives set out in the study, which statistical analyses will be applied (the authors refer to the analyses in the results section).

Results

In the first paragraph of this section (lines 177-183), the authors refer to a table that is not included in the text. We suggest that they list this table (line 177) and present it in the corresponding place in the paper.

Discussion

·         Considering that the aim of the study is to compare the level of efficacy of two therapeutic procedures (Cognitive rehabilitation, Beat Binaural) in the symptomatology associated with sexual hyperactivity and that there are no significant differences in the effects of both interventions, it would be interesting if the authors could provide some commentary on the suitability of applying only one type of therapeutic procedure or the need to combine both to address this problem.

·         Broadening the implications of the results obtained in this study would be another aspect to be considered by the authors.

References

We recommend a general review of the included bibliography as some errors appear. For example,

It is necessary to standardize the presentation of the journals:

·         It is necessary to standardize the presentation of the journals:

o   In some references the journal appears in italics (e.g., line 438), while in others it does not (e.g., line 418).

o   In some references the journal appears in abbreviations (e.g., line 439), while in others it does not (e.g., line 52).

o   Include journal initials in capital letters (e.g., reference 47, line 454).

·         Correct reference number 16 (line 382) as it is repeated twice (references 16 and 18).

Author Response

Dear reviewer, 

thanks a lot. we have included the studies suggested and expanded as requested.

Reviewer 2

To the authors,

 The objective of this research aimed at clarifying the level of efficacy of different therapeutic approaches (Cognitive Rehabilitation and binaural beats) in addressing the symptoms of sexual hyperactivity and associated comorbid conditions, seems to us of interest, because of the important consequences of this problem at different levels (personal, family, health, ...).

In general, the paper is well written, I have only few comments that I hope could be helpful in improving the manuscript.

Response: Dear Reviewer, Thank you so much contributions. We really appreciate it.

Introduction

  • Reading the introduction one confirms the use, in previous literature, of different terms (e.g., Sexual Addiction, Compulsive Sexual Disorder and Hyperactive Sexuality Disorder) to refer to the problem under analysis in this study. Therefore, we suggest that the authors refer to the lack of conceptual clarification in this regard and that they "bet" on a single term throughout the paper, justifying its use. Some articles such as those by Karila, L., Wery, A., Weinstein, A., Cottencin, O., Petit, A., Reynaud, M., & Billieux, J. (2014). Sexual addiction or hypersexual disorder: different terms for the same problem? A review of the literature. Current pharmaceutical design, 20(25), 4012-4020 and the one of Montgomery-Graham, S. (2017). Conceptualization and assessment of hypersexual disorder: A systematic review of the literature. Sexual Medicine Reviews, 5(2), 146-162, may be helpful.

Response: Dear Reviewer. First of all, thank you very much for your suggestions. By adding the studies you mentioned above to the introduction section, we have expanded and improved the introduction section. Added sentences and references are marked in yellow.

  • In line 74, when it is stated that "Research suggests that cognitive impairments may predispose individuals to impulsivity, addiction tendencies, and increased sexual desire," can the authors specify what that research is?

 Response: Dear Reviewer, Thank you so much. We added there references for this sentences

 Materials and Methods

  • This section would be clearer and easier for the potential reader to follow if the authors structured it in different subsections: Procedure, participants, instruments and statistical analysis. Some aspects to consider:

Response: Dear Reviewer, Thank you so much. We used subheadings to make the method section more understandable. We think that the current study is more understandable.

  • Provide more information about the characteristics of the cognitive rehabilitation program that is applied in this research (line 127).

Response: Dear Reviewer, Thank you so much. We added necessary information abour CR program.

  • Include the authors' reference when presenting each of the assessment instruments (e.g., lines 140 and 146)

Response: Dear Reviewer, Thank you so much. In Study Instruments and Questionnaires Section, All measures and questionnaires we used in our study were rewritten and expanded in detail, based on scientific literature.

  • Provide more details about the characteristics of the Carnes Sex Addiction Screening Test (e.g., main dimensions that they assess). In addition, we suggest the inclusion of some items from this questionnaire to exemplify its content.

Response: Dear Reviewer, Thank you so much. We provided more information about the test and marked with yellow. (Line 170-179)

  • Indicate which Likert scale of measurement is used in the Depression, Anxiety, Stress Questionnaire (DASS).

Response: More information were added about the Questionnaire (DASS). Thank you so much. (Line 186-204)

o   Include a section explaining, taking into account the objectives set out in the study, which statistical analyses will be applied (the authors refer to the analyses in the results section).

Response: Dear Reviewer, Thank you so much. We created the which statistical analyses section and necessary information was provided (Line 227-233)

Results

In the first paragraph of this section (lines 177-183), the authors refer to a table that is not included in the text. We suggest that they list this table (line 177) and present it in the corresponding place in the paper.

Response: Dear Reviewer, Thank you so much. The results section was completely changed according to what the other referee wanted. We think the problem is gone.

Discussion

  • Considering that the aim of the study is to compare the level of efficacy of two therapeutic procedures (Cognitive rehabilitation, Beat Binaural) in the symptomatology associated with sexual hyperactivity and that there are no significant differences in the effects of both interventions, it would be interesting if the authors could provide some commentary on the suitability of applying only one type of therapeutic procedure or the need to combine both to address this problem.  -Broadening the implications of the results obtained in this study would be another aspect to be considered by the authors.

Response: The discussion has been revised in the results section. We have also expanded the conclusion section and taken into account your suggestions above and made several suggestions in the last paragraph of the Study's Strengths and Limitations section. Thank you so much.

References

We recommend a general review of the included bibliography as some errors appear. For example,

It is necessary to standardize the presentation of the journals:

  • It is necessary to standardize the presentation of the journals:

o   In some references the journal appears in italics (e.g., line 438), while in others it does not (e.g., line 418).

o   In some references the journal appears in abbreviations (e.g., line 439), while in others it does not (e.g., line 52).

o   Include journal initials in capital letters (e.g., reference 47, line 454).

  • Correct reference number 16 (line 382) as it is repeated twice (references 16 and 18).

Response: Dear Reviewer, Thank you so much. All references were organized. We think the standard has been established.

Reviewer 3 Report

Comments and Suggestions for Authors

This is a very well written manuscript. Below is my feedback.

Introduction:

The introduction is very well written and makes a good case for why the intervention needs to be explored

Methodology:

The methodology is very descriptive and I really appreciate the information the authors have provided in this methodology however, to make it easier for the reader to follow along and be able to look back at it when evaluating the results, I would recommend the authors break the study down into sections. Although the authors can choose to have these sections in any way that they see fit, my recommendation would be to have the following sections: Study design, participants, instruments, procedures, analysis

I also did not read anything about the analyses performed in your methodology. I understand that you put it in your results section however, you need to explicitly state the statistical analysis that you used

Results:

Based on the way you have presented your data in the results section and the analyses that you have stated that you used your analytical techniques are incorrect.

Also it seems that you did not include a table that you refer to in your writing.

The first thing you need to do is perform unpaired t-tests to identify baseline differences between the two groups (if there are baseline differences you need to control for those in your second analysis).

If there are no significant differences between the two groups you'll need to run a 2x2 (2 group x 2 time point) RM-ANOVA, otherwise you will use a 2x2 RM-ANCOVA. Another potential option would be to use a linear mixed-effects modelto control for the baseline differences between the two groups.

Discussion

Based on the analyses I am unable to judge the discussion section.

Author Response

We have made the requested amendments. We performed a rm-ANOVA and redid the analysis. 

Reviewer 3

This is a very well written manuscript. Below is my feedback.

Response: Dear Reviewer, Thank you so much for your contributions.

Introduction:

The introduction is very well written and makes a good case for why the intervention needs to be explored.

Response: Dear Reviewer, Thank you so much for your good comment.

Methodology:

The methodology is very descriptive and I really appreciate the information the authors have provided in this methodology however, to make it easier for the reader to follow along and be able to look back at it when evaluating the results, I would recommend the authors break the study down into sections. Although the authors can choose to have these sections in any way that they see fit, my recommendation would be to have the following sections: Study design, participants, instruments, procedures, analysis.

Response: Response: Dear Reviewer, Thank you so much. We used subheadings to make the method section more understandable. We think that the the method section of the revised study is more understandable.

I also did not read anything about the analyses performed in your methodology. I understand that you put it in your results section however, you need to explicitly state the statistical analysis that you used

 Response: Dear Reviewer, Thank you so much. We created the which statistical analyses section and necessary information was provided (Line 227-233)

Results:

Based on the way you have presented your data in the results section and the analyses that you have stated that you used your analytical techniques are incorrect.

Response: Thank you very much for your constructive contribution. The results section has been completely changed and new analyzes have been performed as per your request.

Also it seems that you did not include a table that you refer to in your writing.

Response: We deleted the all tables. The results were shown through figures. Thank you.

The first thing you need to do is perform unpaired t-tests to identify baseline differences between the two groups (if there are baseline differences you need to control for those in your second analysis).

If there are no significant differences between the two groups you'll need to run a 2x2 (2 group x 2 time point) RM-ANOVA, otherwise you will use a 2x2 RM-ANCOVA. Another potential option would be to use a linear mixed-effects modelto control for the baseline differences between the two groups.

Response: We have made the requested amendments and results section completely changed as compared with the previous one. We performed a rm-ANOVA and redid the analysis. Thank you so much for your contributions.

Discussion

Based on the analyses I am unable to judge the discussion section.

Response: Dear Reviewer, we have improved the discussion section in line with the new results. If you have any suggestions or contributions, we would be pleased. Thank you.

Round 2

Reviewer 3 Report

Comments and Suggestions for Authors

I appreciate the authors addressing my concerns and correcting their analyses. My only minor comment now is if the authors can clearly label the parts of their results sections based on what measure they are presenting it would improve the readibility of the manuscript. 

Author Response

Dear reviewer,

we have clearly labelled the parts of our results sections, as requested, based on what measure we are presenting, to improve the readibility of the manuscript.